# MoaE Is Involved in Response to Oxidative Stress in *Deinococcus radiodurans*

**DOI:** 10.3390/ijms24032441

**Published:** 2023-01-26

**Authors:** Jianling Cai, Maoxu Zhang, Zijing Chen, Ye Zhao, Hong Xu, Bing Tian, Liangyan Wang, Yuejin Hua

**Affiliations:** MOE Key Laboratory of Biosystems Homeostasis and Protection, Institute of Biophysics, College of Life Sciences, Zhejiang University, Hangzhou 310027, China

**Keywords:** *Deinococcus radiodurans*, molybdenum cofactor, MoaE, antioxidation, base analog

## Abstract

Molybdenum ions are covalently bound to molybdenum pterin (MPT) to produce molybdenum cofactor (Moco), a compound essential for the catalytic activity of molybdenum enzymes, which is involved in a variety of biological functions. MoaE is the large subunit of MPT synthase and plays a key role in Moco synthesis. Here, we investigated the function of MoaE in *Deinococcus radiodurans* (DrMoaE) in vitro and in vivo, demonstrating that the protein contributed to the extreme resistance of *D. radiodurans*. The crystal structure of DrMoaE was determined by 1.9 Å resolution. DrMoaE was shown to be a dimer and the dimerization disappeared after Arg110 had been mutated. The deletion of *drmoaE* resulted in sensitivity to DNA damage stress and a slower growth rate in *D. radiodurans*. The increase in *drmoaE* transcript levels the and accumulation of intracellular reactive oxygen species levels under oxidative stress suggested that it was involved in the antioxidant process in *D. radiodurans*. In addition, treatment with the base analog 6-hydroxyaminopurine decreased survival and increased intracellular mutation rates in *drmoaE* deletion mutant strains. Our results reveal that MoaE plays a role in response to external stress mainly through oxidative stress resistance mechanisms in *D. radiodurans*.

## 1. Introduction

The trace element molybdenum is a key component of several enzymes involved in nitrogen, sulfur, and carbon metabolism [1]. Molybdenum usually binds to molybdenum pterin (MPT) to form molybdenum cofactor (Moco) as the catalytic site of molybdoenzymes, such as DMSO reductase family, xanthine oxidase family, and sulfite oxidase family [2]. In all prokaryotes, the biosynthesis of Moco can be divided into four steps: (1) conversion of 5′-GTP into cyclic pyropterin monophosphate (cPMP) [3]; (2) insertion of two sulfur atoms into cPMP to form MPT [4]; (3) intercalation of molybdenum into the disulfide sulfur of MPT, which catalyzes the formation of Mo-MPT [5]; and (4) covalent binding of CMP or GMP to MPT. Finally, Moco is processed by the molybdenum enzyme and catalyzes the redox reaction in which the molybdenum enzyme participates [6].

The conversion of cPMP into MPT in the Moco biosynthetic pathway is responsible for MPT synthase [4]. MPT synthase is a heterotetrameric enzyme composed of two small and two large subunits encoded by *moaD* and *moaE* [7]. Structural analysis of MPT synthases from *Escherichia coli* and *Staphylococcus aureus* have shown that the central dimer is formed by two MoaE subunits, each containing MoaD [4,8], and the small subunit of MPT synthase is thiolated at the C -terminus of MoaD in the form of the carboxylic acid group carrying a sulfur atom [3]. The two MoaE–MoaD dimers function independently during the reaction, with each MoaE subunit bound to one cPMP molecule [9].

In *E. coli* and most organisms, MPT synthase is encoded by the *moaD* and *moaE* genes. In addition to two *moaD* genes (*moaD1* and *moaD2*) and two *moaE* genes (*moaE1* and *moaE2*), *Mycobacterium tuberculosis* also contains an *moaX* gene encoding linearly fused MoaD and MoaE proteins. It has been confirmed that cleavage is required to activate the function of MoaX, the cleavage position is between Gly82 and Ser83 residues, and the Gly81-Gly82 motif is necessary for catalysis [10]. MoaE is essential for Moco synthesis. In *M. tuberculosis*, Moco biosynthesis is also associated with its pathogenicity [11], and the deletion of *moaE* results in the disruption of the Moco biosynthetic pathway [10]. In *Pseudomonas putida*, *moaE* is essential for the generation of the molybdopterin cytosine dinucleotide (Mo-MCD) required for molybdenum enzymes, and its deletion inhibits the growth of the strain in cultures with nicotine as the sole carbon and nitrogen source [12].

*Deinococcus radiodurans* R1 was first isolated from γ-irradiated meat cans in 1956 [13]. In addition to γ irradiation, this robust organism also has high levels of resistance to UV radiation, desiccation, oxidizing agents, and other DNA-damaging agents [14]. *D. radiodurans* possesses strong abilities in preventing the formation of endogenous reactive oxygen species (ROS) [15,16], activating antioxidant defense systems [17,18], as well as removing and degrading damaged bases [19,20]. In *D. radiodurans*, the DrMoaD–MoaE fusion protein is encoded by the *dr_2607* gene, and the JAMM protein encoded by *dr_0402* activates DrMoaD and DrMoaE [21,22]. However, the contribution of the Moco biosynthetic pathway to the extreme resistance of *D. radiodurans* remains unknown.

In the present study, the crystal structure of DrMaoE, an important protein in Moco biosynthesis, was determined, and Arg110 was found to be the key residue for its dimerization. The deletion of *drmoaE* led to an increased susceptibility to stress-induced oxidative damage, a loss of dimethyl sulfoxide (DMSO) reductase activity, and an elevated rate of mutations caused by base analogs. The data above indicate that DrMoaE plays an important role in response to oxidative stress in *D. radiodurans*.

## 2. Results

### 2.1. Overall Structure of DrMoaE

We compared the amino acid sequences of DrMoaE and the homologous proteins from *M. tuberculosis* (MtMoaE), *E. coli* (EcMoaE), and *S. aureus* (SaMoaE), and found that the similarity was high at about 32%, 29.73%, and 37.41%, respectively (Figure 1A).

DrMoaE structural data with a resolution of 1.9 Å were obtained (Appendix A), revealing that each monomeric protein consisted of three α helices and six β sheets. The α/β structure exhibited a hammerhead fold containing an additional four-stranded antiparallel β-sheet domain (Figure 1B,C). To explore the structural similarities and differences between DrMoaE and its homologous proteins, we selected the resolved EcMoaE (PDB: 1NVJ, red) and MtMoaE (PDB: 2WP4, blue) for comparison. The r.m.s.d. values of 1.19 Å and 0.981 Å were comparable with the Cα atoms of EcMoaE and MtMoaE, respectively, suggesting that DrMoaE was similar to its homologous proteins, with a high degree of overlap of the four-stranded antiparallel β sheet, but a displacement of the α1 and α2 helices located around it (Figure 1D). Compared with EcMoaE and MtMoaE, the loop between the β2 and β3 sheets were predominantly disordered, resulting in their apparent deflection (Figure 1D).

### 2.2. Arg110 Is a Key Residue for DrMoaE Dimerization

MoaE can interact with MoaD to complete cPMP processing. The anion pocket of substrate precursor Z (cPMP) bound to EcMoaE was found to consist of His103, Arg104, and Arg39, which corresponded to His109, Arg110, and Arg45 of DrMoaE, respectively. Sequence comparison revealed that these three residues were highly conserved in *D. radiodurans*, *M. tuberculosis*, and *E. coli* (Figure 2A). The His109 and Arg110 amino acid pair of DrMoaE located in the α3 helix was equally capable of forming an anion-binding pocket with the Arg45 of β2 sheet located in another monomer, which matched well with the MtMoaE protein, but exhibited a significant angular shift in *E. coli* (Figure 2B).

In the dimer interaction interface, the two anionic binding pockets in the dotted frames were symmetrical to each other (Figure 2C). In *E. coli*, Arg39 and His103 underwent significant conformational changes before and after substrate binding, while Arg104 remained largely unchanged [23]. It was speculated that EcMoaE Arg104 may function in the dimerization of MoaE in addition to their important role in substrate binding [23]. When the corresponding Arg110 in *D. radiodurans* was mutated to alanine, the gel filtration result of DrMoaE shifted from a single peak of 34 kDa to a single peak of 17 kDa (Figure 2D and Appendix A). The DrMoaE dimer disappeared, suggesting that Arg110 had an irreplaceable role in the stable dimerization formation of DrMoaE.

### 2.3. ΔdrmoaE Mutant Strain Is Sensitive to Damage Stress

To investigate the biological function of *drmoaE*, we analyzed the effects of several DNA-damaging agents on the *ΔdrmoaE* mutant strain. Under H_2_O_2_ treatment, the resistance of the mutant strain significantly decreased by about one or two orders of magnitude compared with the wild-type strain R1, and the complement of *ΔdrmoaE_R110A* could restore some of the growth activity of the *ΔdrmoaE* mutant strain, but it was still about half an order of magnitude lower than that of the R1, whereas the resistance of the *ΔdrmoaE_Cwt* complementary strain was able to fully compensate for the decrease (Figure 3A). When treated with γ-ray or low doses of UV (150 J/m^2^), the mutant did not exhibit a decrease in resistance compared with the wild-type strain (Figure 3B, C). However, the *ΔdrmoaE* mutant exhibited an order of magnitude decrease in survival rate when treated with higher doses of UV (Figure 3B). The deletion of *drmoaE* significantly inhibited the growth of *D. radiodurans*, but the *ΔdrmoaE_R110A* complementary strain could grow at a rate similar to that of wild-type R1 (Figure 3D). These results suggest that *drmoaE* plays an important role in oxidative stress resistance in *D. radiodurans*, while Arg110 is vital in the process.

### 2.4. DrmoaE Contributes to the Antioxidant Process

To establish the role of *drmoaE* in the antioxidant process of *D. radiodurans*, the effect of *drmoaE* deficiency on intracellular ROS scavenging was determined. We compared the accumulation of ROS in the wild-type strain, *ΔdrmoaE* mutant, and *ΔdrmoaE_Cwt* complementary strain and found that the accumulation of ROS in the mutant strain increased significantly compared with the wild-type strain after treatment with H_2_O_2_. Intracellular ROS accumulation in the *ΔdrmoaE* mutant strain was dose-dependent, as the mutant level was about 1.5-fold higher than in the wild-type strain after treatment with 80 mM H_2_O_2_ (Figure 4A). These results indicate that *drmoaE* is involved in the antioxidant process of *D. radiodurans* by affecting intracellular ROS accumulation.

The transcript levels of *drmoaE* under oxidative stress were further analyzed using reverse transcription polymerase chain reaction. The wild-type strain was treated with 40 mM H_2_O_2_ for 0, 15, 30, and 60 min, and the transcript levels of *drmoaE* increased with longer treatment time, reaching the highest level at 30 min (Figure 4B). This indicates that *drmoaE* participates in the antioxidant process of *D. radiodurans* by increasing its transcript levels.

### 2.5. DMSO Reductase Activity Is Decreased in the ΔdrmoaE Mutant Strain

Superoxide dismutase (SOD) and catalase (CAT) are important components of the antioxidant system. Under normal physiological conditions, the activity of SOD and CAT in *D. radiodurans* is approximately 6- and 20-fold higher, respectively, than that of *E. coli*. We compared the intracellular SOD and CAT activities of the wild-type strain, *ΔdrmoaE* mutant, and *ΔdrmoaE_Cwt* complementary strain, and found that the deletion of *drmoaE* did not significantly decrease the activities of SOD and CAT in vivo (Figure 5A,B), indicating that *drmoaE* does not respond to oxidative stress by affecting SOD and CAT activities.

It has been shown that the activities of some Moco-dependent oxidoreductases, such as DMSO reductase, depend on Moco biosynthesis [24]. The DMSO reductase family found in bacteria and archaea catalyzes oxygen atom transfer and simple redox reactions. The intracellular DMSO reductase activity of the wild-type strain, *ΔdrmoaE* mutant strain, and *ΔdrmoaE_Cwt* complementary strain was analyzed, and intracellular DMSO reductase activity was found to decrease significantly when *drmoaE* was absent, until it was undetectable (Figure 5C). This suggests that DrMoaE responds to antioxidant processes by affecting DMSO reductase activity.

### 2.6. DrMoaE Responds to Base Analog Damage

Our above results show that the *ΔdrmoaE* mutant strain was sensitive to H_2_O_2_ and high-dose UV, implying that *drmoaE* plays a role in the cellular defense mechanism against oxidative damage to DNA. In *E. coli*, the deletion of Moco-related genes, such as *moeA* results in DNA mutation, were induced by base analog 6-hydroxyaminopurine (HAP) [25]. In *Salmonella*, mutations caused by the deletion of Moco-related genes in addition to HAP were also affected by other base analogs other than HAP, such as hydroxylamine aminopurine [26]. The survival rate of the *ΔdrmoaE* mutant strain in the presence of HAP was decreased to about 50% relative to the wild-type and *ΔdrmoaE_Cwt* complementary strains (Figure 6A). Screening using TGY solid medium containing 50 μg/mL rifampicin revealed that the deletion of *drmoaE* resulted in a rapid increase in mutation frequency eight-fold that of the wild-type strain, whereas the mutation rate of the wild-type and *ΔdrmoaE_Cwt* complementary strains did not change significantly (Figure 6B). High doses of UV and γ rays also broke down water molecules, causing some oxidative damage and producing base analogs. Therefore, one of the main functions of *drmoaE* in the antioxidant process may be to metabolize the base analogs produced by oxidative stress and reduce protein and DNA damage under oxidative damage conditions, thus accelerating the recovery of the strains from various external stresses.

## 3. Discussion

MPT synthase catalyzes the conversion of precursor Z (cPMP) into MPT via sulfur transfer of MoaD thiocarboxylate [11]. The mechanism of MPT synthase relies heavily on the precise interaction between the active form of MoaD carrying the thiocarboxylate and the MoaE protein. In the present study, we solved the crystal structure of DrMoaE by 1.9 Å resolution. Compared with previously reported structures of homologous proteins from *E. coli* and *M. tuberculosis*, DrMoaE had a high similarity but exhibited a significant angular deflection of the loop sequence between its β2 and β3 sheets. The positively charged residues His109, Arg110, and Arg45, which can form an anion-binding pocket in *D. radiodurans*, were the most conserved in different organisms. However, a significant angular shift in the positively charged residues of the substrate-binding pocket of EcMoaE compared to DrMoaE was found in the structural comparison, suggesting that there may be variability in the substrate-binding mode in different species. Meanwhile, the structural analysis revealed that Arg110 is essential for the dimerization of DrMoaE, and the stable dimerization state of DrMoaE disappeared after mutation of this site, indicating that Arg110 is important in the dimerization of DrMoaE.

The *ΔdrmoaE* mutant strain was sensitive to various DNA-damaging agents, especially to oxidative stress caused by H_2_O_2_. The deletion of *drmoaE* resulted in growth restriction of *D. radiodurans*. Intracellular ROS accumulation confirmed that *drmoaE* deficiency led to ROS accumulation under oxidative stress in *D. radiodurans*, while the transcript levels of *drmoaE* in the wild-type strain were elevated under oxidative stress. The above results suggest that *drmoaE* plays an important role in oxidative stress resistance in *D. radiodurans*.

CAT and SOD are important ROS scavengers in *D. radiodurans*, and the absence of *drmoaE* has little effect on their activities, suggesting that an additional effective ROS scavenging mechanism is required for DrMoaE to perform its function. It was shown that the ratio of MoO_4_^2−^ affected the capacity of redox metabolic in *Agave* [27]. The transport pathway of MoO_4_^2−^ is mainly converted into Moco through the Moco biosynthesis pathway to complete the function of a series of molybdenum enzymes such as DMSO reductase. The loss of DMSO reductase activity in the *ΔdrmoaE* mutant suggests that the function of molybdenum enzymes is hindered by the absence of *drmoaE*, thereby affecting the antioxidant process in which molybdenum enzymes are involved in *D. radiodurans*.

Modified nucleobases that can participate in cellular processes together with natural substrates which are toxic and/or mutagenic are traditionally referred to as base analogs [26]. In organisms, base analogs can act as products of oxidative stress damage, such as 5-hydroxy-dCTP [28], 2-hydroxy-dATP [29], and 8-oxo-dGTP [30]. Base analogs are potent mutagens, such as HAP for adenine analogs, which are inserted into the DNA strand by DNA polymerase, leading to incorrect DNA replication and mutations, and are therefore commonly used to test the metabolism of nucleotide precursors or the mechanisms of DNA replication and repair [31]. Our study showed that DNA and DNA bases were exposed to peroxyl radicals (ROO^−^), a major intracellular oxidant and oxidative stress product, with HAP being the major product [32]. The *ΔdrmoaE* mutant strain exhibited a significant decrease in survival rate and a significant increase in mutation rate in vivo after treatment with HAP, suggesting that the deletion of *drmoaE* leads to the accumulation of base analogs in vivo and an increase in DNA mutations. In *E. coli*, the DMSO reductase-dependent deletion of the *bisC* Moco pathway is insensitive to HAP [33]. In vitro studies have found that mammalian xanthine oxidase is able to reduce HAP to adenine [34]. The treatment of wild-type strains with febuxostat, a specific inhibitor of xanthine dehydrogenase and xanthine oxidase, was to found to have no effect on the mutation rate in *D. radiodurans*, nor did hypoxanthine-specific inhibition of xanthine dehydrogenase and xanthine oxidase in *E. coli*. However, cell lysates from *E. coli moaE* deletion mutant strains resulted in a decrease in the conversion of HAP to adenine with reduced efficiency [35], suggesting that *moaE* mediates a specific molybdenum enzyme pathway involved in the metabolism of base analogs in bacteria, other than the common molybdenum enzyme pathway reported to date.

In conclusion, our results support that DrMoaE possesses a conserved MoaE protein structure, with a more disordered loop between the β2 and β3 sheets, while the cPMP-binding pocket Arg110 residue is a key site for its dimerization, which responds to the oxidative damage process in *D. radiodurans*, and is thus able to reduce base damage and genomic DNA mutations to defend against various external stresses.

## 4. Materials and Methods

### 4.1. Strains and Growth Conditions

All strains, plasmids, and primers used in this study are listed in Appendix A. *E. coli* strains were grown in Luria-Bertani (LB) liquid medium (1% tryptone, 0.5% yeast extract, and 1% sodium chloride) or on agar (1.5% Bacto-agar) plates supplemented with appropriate antibiotics at 37 °C. All *D. radiodurans* strains were grown at 30 °C in tryptone glucose yeast extract (TGY) liquid media or on agar plates (0.5% tryptone, 0.1% glucose, and 0.3% yeast extract) supplemented with appropriate antibiotics.

### 4.2. Expression and Purification of Proteins

The *drmoaE* gene was amplified, digested using *Nde*I (upstream) and *BamH*I (downstream), and ligated to the pET28a expression vector which has been digested using *Nde*I and *BamH*I. The targeted mutant fragments were obtained by PCR amplification of selected mutant sites. Primers were shown in Appendix A. The successfully constructed plasmid was transformed into *E. coli* BL21 (DE3) and induced in LB liquid medium containing 40 μg/mL kanamycin and 0.4 mM isopropyl-β-D-thiogalactopyranoside (IPTG) for 5 h at 30 °C. Cells were harvested and resuspended in lysis buffer (20 mM Tris-HCl, pH 8.0; 200 mM NaCl; 5% (*w*/*v*) glycerol; 3 mM β-mercaptoethanol) followed by sonication. The bacterial lysates were centrifuged at 15,000 rpm for 30 min at 4 °C, and the supernatant was filtered through a membrane and purified using a Ni-NTA column (1 mL, GE Healthcare Biosciences, Boston, Massachusetts, USA) equilibrated with buffer A (20 mM Tris-HCl pH 8.0; 200 mM NaCl; 5% (*w*/*v*) glycerol), and washed with buffer B (20 mM Tris-HCl pH 8.0; 200 mM NaCl; 5% (*w*/*v*) glycerol; 300 mM imidazole) to separate impurities from target proteins. Next, protein samples were purified using a HiTrap Q ion exchange column (GE Healthcare Biosciences, Boston, Massachusetts, USA) equilibrated with buffer A and washed with buffer C (20 mM Tris-HCl pH 8.0; 1 M NaCl; 5% (*w*/*v*) glycerol). Finally, protein purification was performed using a Superdex75 10/300 GL (GE Healthcare Biosciences, Boston, Massachusetts, USA) equilibrated with buffer D (20 mM Tris-HCl; pH 8.0; 200 mM KCl; 1 mM DTT), followed by protein concentration using a concentration centrifuge tube.

### 4.3. Crystallization, Data Collection, and Structure Determination

Purified fresh DrMoaE protein (N-terminal with 6×His) was concentrated to 2.8 mg/mL and screened for DrMoaE crystallization conditions using a protein crystallization kit (HR-114) incubated at 293 K. About 1 week later, DrMoaE crystals were obtained at a reservoir condition containing 0.2 M lithium sulfate, 0.1 M Tris pH 8.5, and 22% PEG3350. To the microwells with DrMoaE protein crystals, 20% cryoprotectant (0.2 M lithium sulfate, 0.1 M Tris pH 8.5, 22% PEG3350, and 20% (*w*/*v*) glycerol) were added and left for 3 min, and the retrieved protein crystals were stored in liquid nitrogen. DrMoaE protein crystals were subjected to X-ray diffraction at the beamline BL17U of the Shanghai Light Source (SSRF) and were integrated and scaled using the XDS suite suit [36]. The structure of DrMoaE was solved by molecular replacement using *E. coli* homolog (PDB:1NVJ; [23]) as the search model. The refinement of the structure was performed using PHENIX [37], and manual modeling was performed using COOT [38]. The refined structure included 139 amino acids of DrMoaE (residues 9–145 of each monomer), and the structure of DrMoaE was mainly presented using PyMOL.

### 4.4. Construction of Mutant and Complementary Strains

The deletion mutant strains were constructed using the previously described triple ligation method [39]. Briefly, the amplified upstream and downstream fragments of the target gene were digested using *Hind*III and *BamH*I, respectively, and ligated to the streptavidin gene that had been digested using *Hind*III and *BamH*I. The ligation product was transformed into *D. radiodurans*. Then, the deletion mutant strain was screened using a TGY medium containing 10 μg/mL streptomycin. The *drmoaE* and *drmoaER110A* were cloned into the pRADK vector with the *groEL* promoter of *D. radiodurans* for complementary strain construction. The constructed plasmid was transformed into the *ΔdrmoaE* mutant strain to obtain the *ΔdrmoaE_Cwt* complementary strain and *ΔdrmoaE_R110A* complementary strain.

### 4.5. Phenotypic and Growth Curve Analysis

Both the phenotypic analysis and the growth curve were determined as previously described [40]. The wild-type strain, *ΔdrmoaE* mutant strain, and *ΔdrmoaE_Cwt* complementary strain were cultured to OD_600_ = 1.0. For H_2_O_2_ treatment, strains were treated with 20–60 mM H_2_O_2_ for 30 min and diluted 10^−4^ times using sterile PBS solution, and 100 μL of the diluted bacterial solution was applied to the TGY plates for 2–3 days at 30 °C for counting. For UV treatment, strains were diluted 10^−4^ times using sterile PBS solution, and 100 μL of the diluted bacterial solution was applied to TGY solid medium and then treated with 0–600 J/m^2^ UV and incubated at 30 °C for 2–3 days. For γ-ray treatment, the strains were irradiated with 0–8 kGy γ-rays (^60^Co, Zhejiang Academy of Agricultural Sciences, China) and diluted 10^−4^ times using sterile PBS solution, and 100 μL of the diluted bacterial solution was applied to the in TGY plates for 2–3 days at 30 °C for counting. For the growth curve, the wild-type strain, *ΔdrmoaE* mutant strain, and *ΔdrmoaE_Cwt* complementary strain were grown to OD_600_ = 1.0 and 100 μL was transferred to 5 mL of TGY medium without antibiotics, and the OD_600_ value of each strain was detected every 2 h.

### 4.6. Detection of Intracellular ROS Content

Intracellular ROS levels were detected by a molecular probe, 2′,7′-dichlorofluorescein diacetate (DCFH-DA), as previously described [41]. Briefly, wild-type strains, *ΔdrmoaE* mutant strains, and *ΔdrmoaE_Cwt* complementary strains were grown to OD_600_ = 1.0, the supernatant was removed by centrifugation at 5000 rpm for 3 min and washed 3 times with PBS buffer. The organisms were incubated with DCFH-DA for 30 min at 37 °C. After incubation, the cells were washed 3 times with PBS buffer and resuspended in 2 mL of PBS buffer. Then, 1 mL of the sample was treated with 0, 40, and 80 mM H_2_O_2_ for 30 min, respectively. The fluorescence intensity (excitation wavelength 488 nm, detection wavelength 525 nm) was measured using a fluorescence spectrophotometer (SpectraMax M5, Molecular Devices, Sunnyvale, California, USA).

### 4.7. Real-Time Quantitative PCR (RT-qPCR)

The expression of *drmoaE* under oxidative stress was analyzed by real-time quantitative PCR (RT-qPCR) as described previously [42]. Wild-type strains were incubated to OD_600_ = 1.0, treated with 40 mM H_2_O_2_ for 30 min, and the reaction was terminated using catalase. Total RNA was extracted using TRIZOL reagent (Invitrogen, Carlsbad, CA, USA), and RT-qPCR analysis was performed using SYBR Premix Ex Taq (TaKaRa Biotechnology, Kyoto Prefecture, Japan) to collect data. Primers are shown in Appendix A. Data were collected and differences in relative transcript abundance levels were calculated. The internal control was the *dr_1343* gene encoding glyceraldehyde 3-phosphate dehydrogenase (GADPH).

### 4.8. Analysis of Superoxide Dismutase (SOD) Activity

Superoxide dismutase (SOD) activity was determined using the nitrogen blue tetrazolium (NBT) method. The wild-type strain, *ΔdrmoaE* mutant strain, and *ΔdrmoaE_Cwt* complementary strain were grown to OD_600_ = 1.0, and 1 mL of bacterial solution was centrifuged at room temperature for 3 min at 5000 rpm to remove the supernatant. After that, an equal volume of extract was added to resuspend the bacteria, and the bacteria were completely crushed in an ice bath using an ultrasonic crusher. The absorbance value at 560 nm was measured using a spectrophotometer, following which cell lysates were incubated with the reaction reagent of Superoxide dismutase (SOD) activity detection kit (Beijing Solarbio Science & Technology, Beijing, China) in a water bath at 37 °C for 30 min. Superoxide anion (O_2_^−^) that was produced by the reaction of xanthine and xanthine oxidase in the kit reduced nitro-blue tetrazolium to blue formazan. After the addition of SOD, O_2_^−^ was cleared so as to inhibit the formation of formazan. It was considered an enzyme activity unit (U) when the percentage of SOD enzyme inhibition of the xanthine oxidase coupling reaction system described above was 50% at 37 °C.

### 4.9. Analysis of Catalase Activity

Under sufficient hydrogen peroxide conditions, catalase can catalyze H_2_O_2_ to produce water and oxygen completely, and the residual H_2_O_2_ can be oxidized by extra catalase to produce the red product (N-(4-antipyrine)-3-chloro-5-sulfonate-p-benzoquinonemonoimine), which can be detected at the wavelength of 520 nm. The wild-type strain, *ΔdrmoaE* mutant strain, and *ΔdrmoaE_Cwt* complementary strain were grown to OD_600_ = 1.0, and the supernatant was removed by centrifugation at 5000 rpm for 3 min at room temperature, and the bacteria were resuspended using lysate and incubated at 37 °C for 30 min for lysis. The cell lysis supernatant, hydrogen peroxide assay buffer, and 250 mM hydrogen peroxide solution were added in a total reaction volume of 50 μL at 25 °C for 5 min, and the reaction was terminated using 450 μL of peroxidase reaction termination solution. The absorbance value at 520 nm was measured after incubation with the color development solution at 25 °C for 15 min. Catalase activity was measured using the Catalase Assay Kit (Beyotime Biotechnology, Shanghai, China). The unit (U) indicated that one enzyme activity unit (1 unit) catalyzes the breakdown of 1 μmol H_2_O_2_ in 1 min at 25 °C, pH 7.0.

### 4.10. Analysis of DMSO Reductase Activity

DMSO reductase activity was assayed as described previously [43]. Briefly, wild-type strain, *ΔdrmoaE* mutant strain, and *ΔdrmoaE_Cwt* complementary strain were cultured to OD_600_ = 1.0, harvested by centrifugation, washed using buffer A (50 mM Tris-HCl, pH 7.5; 1 mM EDTA, pH 8.0; 2 M NaCl), and resuspended in 20 mL of buffer A. Cells were lysed using a high-pressure disruptor, and supernatants were collected by centrifugation at 4 °C, 15,000 rpm for 30 min, and assayed for total protein concentration using the Bradford Protein Assay Kit (Beyotime Biotechnology, Shanghai, China). Cell lysates (1–1.5 mg protein), buffer A, and 0.3 mM methyl violet crystal were mixed and reacted in an anaerobic environment. The system was adjusted to 1–1.2 at OD_600_ absorbance value with fresh Na_2_S_2_O_4_ (20 mM) and NaHCO_3_ (20 mM), followed by the addition of DMSO (10 mM) for the assay. The reaction was assayed for a total of 3.5 min at 15 s intervals. Each DMSO reductase activity unit (U) was defined as 1 μmol of substrate consumed per minute at room temperature. The extinction coefficient of methyl viologen A_600_ nm was 13.6 (mM^−1^·cm^−1^).

### 4.11. 6-Hydroxyaminopurine (HAP)-Mediated Growth Inhibition

The wild-type strain, *ΔdrmoaE* mutant strain, and *ΔdrmoaE_Cwt* complementary strain cultured to OD_600_ = 1.0 were divided into 1 mL aliquots into 1.5 mL centrifuge tubes, centrifuged at 5000 rpm for 3 min to remove the supernatant, and resuspended using 1 mL sterile PBS buffer. After that, an appropriate amount of 6-hydroxyaminopurine (HAP) solution (mother liquor concentration 0.1 mg/mL) was added to the resuspension solution and incubated at 30 °C for 2 h, and then kept on ice. The bacterial solution was diluted 10^4^ times using sterile PBS buffer, and 100 μL of the dilution was applied to the plate on TGY solid medium and incubated at 30 °C for 2 days. We counted the number of surviving cells in media additives with HAP or not and calculated the percentage of surviving cells [35].

### 4.12. Mutant Frequency Determinations

The wild-type strain, *ΔdrmoaE* mutant strain, and *ΔdrmoaE_Cwt* complementary strain cultured to OD_600_ = 0.4 were divided into 1 mL aliquots and into 1.5 mL centrifuge tubes with appropriate amounts of HAP solution and incubated overnight at 30 °C. A total of 100 μL of the bacterial solution was applied to the plate on a TGY solid medium containing 50 μg/mL of rifampicin in order to obtain the number of Rif^r^ cells per strain. The bacterial solution was diluted using sterile PBS buffer and 100 μL of the bacterial solution dilution was applied to the plates without rifampicin to obtain the total number of cells per strain. The number of Rif^r^ cells divided by the total number of cells was used to determine the mutation frequency [35].

## Figures and Tables

**Figure 1 ijms-24-02441-f001:**
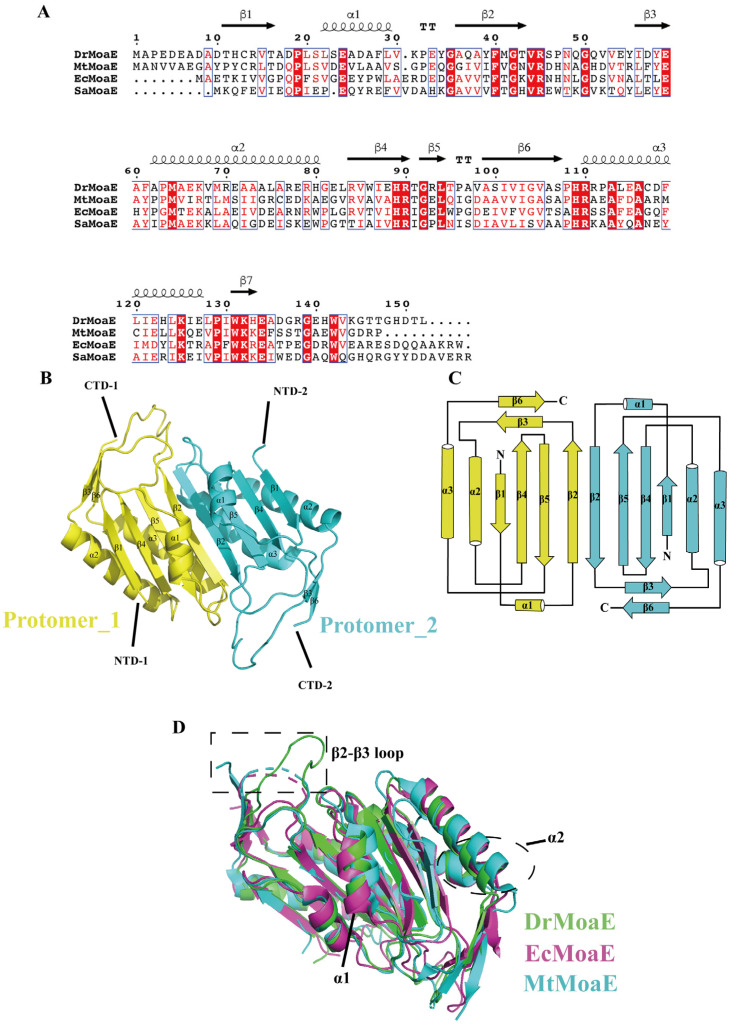
Overall structure of DrMoaE. (**A**) Sequence alignment of DrMoaE with homologous proteins of *M. tuberculosis* (MtMoaE), *E. coli* (EcMoaE), and *S. aureus* (SaMoaE). (**B**) The dimer of DrMoaE is shown as a cartoon and the two protomers are represented in yellow and blue, respectively. Each protomer’s α-helices, β-sheet, N-terminal (NTD), and C-terminal (CTD) are labeled. (**C**) Topology diagram of DrMoaE. The color is the same as in panel B. The N-terminal (N) and C-terminal (C) of each protomer are labeled. (**D**) Structural alignment of MoaE. *D. radiodurans* (green), *E. coli* (red), *M. tuberculosis* (blue). The dotted box indicates the loop between β2 and β3 sheets.

**Figure 2 ijms-24-02441-f002:**
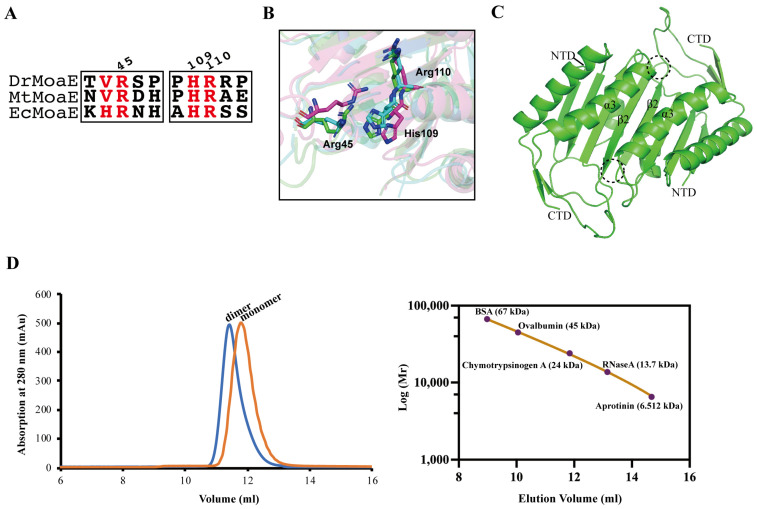
Roles of key amino acid residues in DrMoaE. (**A**) Sequence alignment of DrMoaE, MtMoaE, and EcMoaE. (**B**) Structure alignment of His109, Arg110, and Arg45 of DrMoaE with the corresponding sites of EcMoaE and MtMoaE. *D. radiodurans* (green), *E. coli* (purple), *M. tuberculosis* (blue). (**C**) The structure of DrMoaE; the dotted box represents the anion binding pocket. (**D**) Gel filtration profiles of DrMoaE and DrMoaER110A. Blue represents DrMoaE and orange represents DrMoaER110A. The protein standard curve was generated using a mixture of aprotinin (6.512 kDa), RNaseA (13.7 kDa), chymotrypsinogen A (24 kDa), ovalbumin (45 kDa), and BSA (67 kDa).

**Figure 3 ijms-24-02441-f003:**
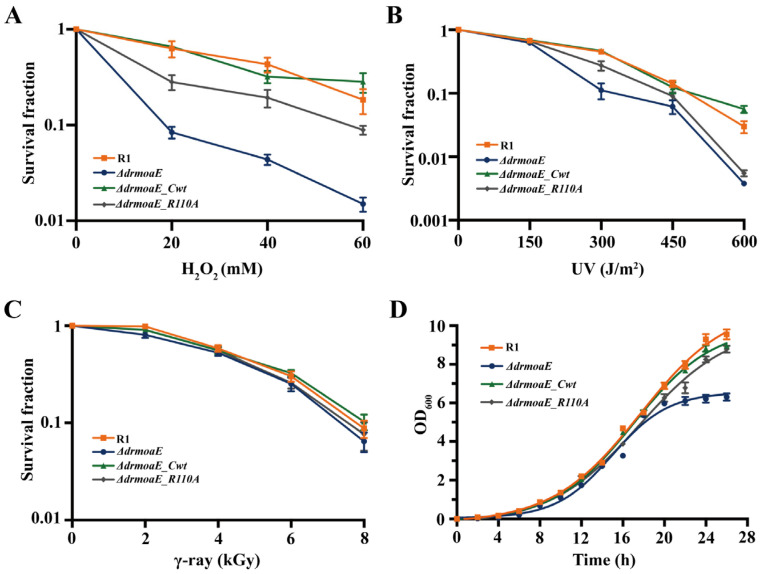
Phenotype and growth analysis of *drmoaE*. (**A**) Survival curves of the wild-type strain (R1), the mutant strain (*ΔdrmoaE*), and the complementary strain (*ΔdrmoaE_Cwt* and *ΔdrmoaE_R110A*) under H_2_O_2_ (0–60 mM), respectively. (**B**) Survival curves of the wild-type strain (R1), the mutant strain (*ΔdrmoaE*), and the complementary strain (*ΔdrmoaE_Cwt* and *ΔdrmoaE_R110A*) under UV (0–600 J/m^2^), respectively. (**C**) Survival curves of the wild-type strain (R1), the mutant strain (*ΔdrmoaE*), and the complementary strain (*ΔdrmoaE_Cwt* and *ΔdrmoaE_R110A*) under γ-ray (0–8 kGy), respectively. (**D**) Growth curve of the wild-type strain (R1), mutant strain (*ΔdrmoaE*), and complementary strain (*ΔdrmoaE_Cwt* and *ΔdrmoaE_R110A*). Each value represents the mean of three assays (mean ± SD).

**Figure 4 ijms-24-02441-f004:**
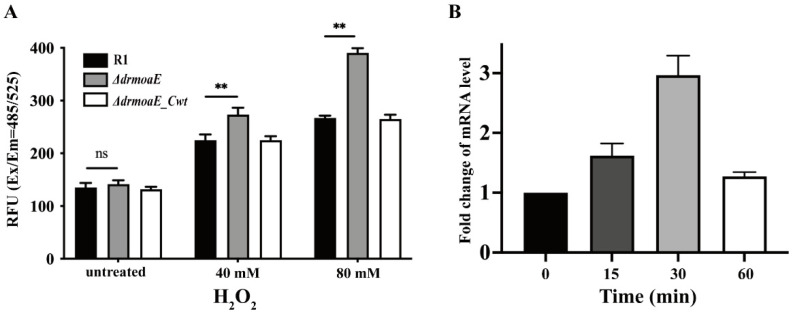
Antioxidant function of DrMoaE. (**A**) The level of ROS accumulation in cells after 0, 40, and 80 mM H_2_O_2_ treatment, respectively. “Untreated” represents a concentration of 0 mM H_2_O_2_. RFU means relative fluorescence units. Each value represents the mean of three assays (mean ± SD). “ns” means not significant. ** *p* < 0.01. (**B**) The mRNA levels of *drmoaE* after exposure to 40 mM H_2_O_2_ for 0, 15, 30, and 60 min. Each value represents the mean of three assays (mean ± SD).

**Figure 5 ijms-24-02441-f005:**
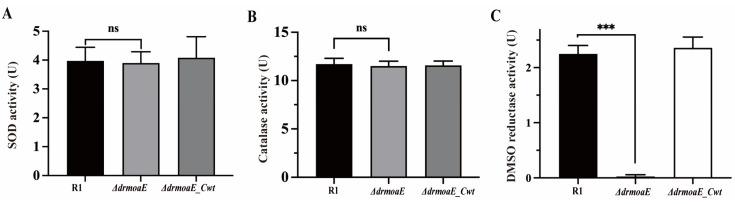
Activities of superoxide dismutase (SOD) (**A**), catalase (CAT) (**B**), and DMSO reductase (**C**) in the wild-type strain (R1), the mutant strain (*ΔdrmoaE*), and complementary strain (*ΔdrmoaE_Cwt*). Each value represents the mean of three assays (mean ± SD). “ns” means not significant. *** *p* < 0.001.

**Figure 6 ijms-24-02441-f006:**
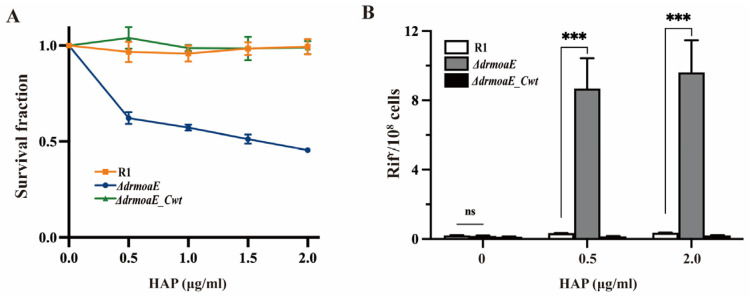
Survival rate (**A**) and mutation rate (**B**) of the wild-type strain (R1), the mutant strain (*ΔdrmoaE*), and the complementary strain (*ΔdrmoaE_Cwt*) exposed to HAP. The survival rate was determined after a 2 h exposure to HAP in the TGY medium. The mutation rate was measured after overnight exposure to HAP in the TGY medium. Each value represents the mean of three assays (mean ± SD). “ns” means not significant. *** *p* < 0.001.

## Data Availability

The coordinates and structure factors have been deposited to Protein Data Bank with accession codes 8HLG.

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
