# Peer review of "MoaE Is Involved in Response to Oxidative Stress in Deinococcus radiodurans"

_ijms, 2023, doi:10.3390/ijms24032441_

Round 1

Reviewer 1 Report

This manuscript entitled "MoaE is involved in response to oxidative stress in Deinococcus radiodurans” written by Cai et al. describes the crystal structure of D. radiodurans MoaE protein with 1.9 AÌŠ resolution. DrMoaE structure is similar with that of EcMoaE and MtMoaE while the loop between β2–β3 sheets is relatively disordered. Mutant studies revealed that DrMoaE is responsible for the oxidative stress resistance of D. radiodurans by accompaning with DMSO reductase activity, base analog mediated accumulation of mutation. This study is informative for the biochemical and physiological function of MoaE protein and related field. The methodology used and the result are proper and concise. Overall, I think this manuscript is possible to be published in International Journal of Molecular Sciences with minor changes.

Minor comments

1) Lines 14-15

- The crystal structure of DrMoaE was obtained, and its resolution was determined by diffraction to be 1.9 AÌŠ

-> The crystal structure of DrMoaE was determined by 1.9 AÌŠ resolution.

2) Line 22

- through oxidative resistance mechanisms in D. radiodurans.

-> through oxidative stress resistance mechanisms in D. radiodurans.

3) Line 28

- as the catalytic site of molybdenase.

-> Please verify what molybdenase means? It seems better to use the common name of proteins authors intend to mention.

4) Line 51

- for generation of the Mo-MCD

-> Please give a full description of MCD.

5) Line 70

- 2.1. Overall crystal structure of DrMoaE

-> I suggest to change this title as ‘2.1. Overall structure of DrMoaE.

6) Lines 72-73

- from Mycobacterium tuberculosis (MtMoaE), Escherichia coli (EcMoaE), and Staphylococcus aureus (SaMoaE),

-> from M. tuberculosis (MtMoaE), E. coli (EcMoaE), and S. aureus (SaMoaE),

7) Line 76-77

- six β folds.

-> six β sheets.  

8) Line 84

- but a displacement of the α2 and α3 helices located around it.

-> In Figure 1D, it is not clear to show whether these two helices exhibit displacement to EcMoaE and MtMoaE. Please consider to clarify this either by changing Fig. 1D or give any numerical parameters in the text.

9) lines 85-86

- the loop sequence residues located between the β2 and β3 folds around DrMoaE were predominantly disordered

-> the loop between β2 and β3 were predominantly disordered

10) Line 92

- the same as in panel A.

-> the same as in panel B.

11) Lines 92-94

-> It seems that these sentences include several incomplete writings. Please correct this.

-> each protomer are labe ral (?) alignments of MoaE

-> indicates the β2led (?). (D) Structure-β3 loop (?). 

12) Lines 114-115

- with homologous sites of EcMoaE and MtMoaE.

-> with the corresponding sites of EcMoaE and MtMoaE.

13) Lines 116-117

- Dimerization of DrMoaE and DrMoaER110A.

-> Suggestion: Gel filtration profiles of DrMoaE and DrMoaER110A.

-> Please give the calculated MW information with MW standard proteins.

14) Throughout the manuscript, please use complementary or complementation instead of compensatory or compensation.

15) Line 130

- in oxidation resistance

-> in oxidative stress resistance

16) Figure 5 and Figure 6

-> Figures 5, 6 and its legends are not matched. Please clarify this and correct adequately.

17) Line 183

- by affecting molybdenum enzyme activity.

-> Suggestion: by affecting DMSO reductase activity.

18) Lines 212-213

- In the present study, structural data of DrMoaE protein with a resolution of 1.9 Å were obtained using the diffraction technique.

-> In the present study, we solved the crystal structure of DrMoaE by 1.9 Å resolution.

19) Throughout the manuscript, please clarify that ‘β2–β3 folds’ is proper term to express what authors mean.

20) Line 273

- All D. r strains were

-> All D. radiodurans strains were

21) Line 299-300

- at pool conditions

-> at a reservoir condition

22) Line 301

- 20% antifreeze

-> 20% cryoprotectant (please give detailed component)

23) Line 303

- at the BL17U line station of

-> at the beamline BL17U of

24) Line 305-309

- and modeled by molecular replacement using E. coli homologous structure MoaE (PDB:1NVJ). The correction of the structure was carried out by PHENIX software, and manual modeling was performed using COOT software. The refined structure included 139 amino acids of DrMoaE (residues 9-145 of each monomer), and the structure of DrMoaE was mainly presented using PyMOL software.

-> (reference for XDS program suit) and molecular replacement phasing using EcMoaE (PDB:1NVJ) as a search model was performed by using ~~(give MR program used in this study and its reference). Manual model building was performed using COOT software (add reference). The refined structure included 139 amino acids of DrMoaE (residues 9-145 of each monomer), and the structure of DrMoaE was mainly presented using PyMOL software.

25) Throughout the manuscript, please unify mL or ml.

26) Lines 365-367

- One SOD enzyme activity unit (U) was defined when the inhibition percent of the reaction system was 50%.

-> For arbitrary unit definition, authors need to give more information such as the concentration of substrate, temperature, and pH and so on.

Author Response

Point 1: Lines 14-15

- The crystal structure of DrMoaE was obtained, and its resolution was determined by diffraction to be 1.9 AÌŠ

-> The crystal structure of DrMoaE was determined by 1.9 AÌŠ resolution.

Response 1: Thank you for your comment. It has been corrected (Line 14, Page 1).

Point 2: Line 22

- through oxidative resistance mechanisms in D. radiodurans.

-> through oxidative stress resistance mechanisms in D. radiodurans.

Response 2: Thank you. It has been corrected (Lines 22-23, Page 1).

Point 3: Line 28

- as the catalytic site of molybdenase.

-> Please verify what molybdenase means? It seems better to use the common name of proteins authors intend to mention.

Response 3: Thank you for your suggestion. It has been improved as “Molybdenum usually binds to molybdenum pterin (MPT) to form molybdenum cofactor (Moco) as the catalytic site of molybdoenzymes, such as DMSO reductase family, xanthine oxidase family, and sulfite oxidase family” (Lines 29-30, Page 1).

 Point 4: Line 51

- for generation of the Mo-MCD

-> Please give a full description of MCD.

Response 4: Thank you for your reminder. It has been revised as “molybdopterin cytosine dinucleotide (Mo-MCD)” (Line 54, Page 2).

Point 5: Line 70

- 2.1. Overall crystal structure of DrMoaE

-> I suggest to change this title as ‘2.1. Overall structure of DrMoaE.

Response 5: Thank you. It has been improved (Line 73, Page 2).

Point 6: Lines 72-73

- from Mycobacterium tuberculosis (MtMoaE), Escherichia coli (EcMoaE), and Staphylococcus aureus (SaMoaE),

-> from M. tuberculosis (MtMoaE), E. coli (EcMoaE), and S. aureus (SaMoaE),

Response 6: Thank you. It has been corrected (Line 75, Page 2).

 Point 7: Line 76-77

- six β folds.

-> six β sheets.

Response 7: Thank you. It has been corrected (Lines 81, 82, and 89, Page 2).

 Point 8: Line 84

- but a displacement of the α2 and α3 helices located around it.

-> In Figure 1D, it is not clear to show whether these two helices exhibit displacement to EcMoaE and MtMoaE. Please consider to clarify this either by changing Fig. 1D or give any numerical parameters in the text.

Response 8: Thank you. Based on your suggestion, we have revisited the structure of DrMoaE and found that α1 and α2 have a slight shift, which has been modified and shown in Figure 1D.

Point 9: lines 85-86

- the loop sequence residues located between the β2 and β3 folds around DrMoaE were predominantly disordered

-> the loop between β2 and β3 were predominantly disordered

Response 9: Thank you. It has been corrected (Line 89, Page 2).

 Point 10: Line 92

- the same as in panel A.

-> the same as in panel B.

Response 10: Thank you. It has been corrected (Line 98, Page 3).

 Point 11: Lines 92-94

-> It seems that these sentences include several incomplete writings. Please correct this.

-> each protomer are labe ral (?) alignments of MoaE

-> indicates the β2led (?). (D) Structure-β3 loop (?).

Response 11: Thank you for the reminder. It has been revised as “(C) Topology diagram of DrMoaE. The color is the same as in panel B. The N-terminal (N) and C-terminal (C) of each protomer are labeled. (D) Structural alignment of MoaE. D. radiodurans (green), E. coli (red), M. tuberculosis (blue). The dotted box indicates the loop between β2 and β3 sheets” (Lines 97-102, Page 3).

 Point 12: Lines 114-115

- with homologous sites of EcMoaE and MtMoaE.

-> with the corresponding sites of EcMoaE and MtMoaE.

Response 12: Thank you. It has been corrected (Line 124-125, Page 4).

 Point 13: Lines 116-117

- Dimerization of DrMoaE and DrMoaER110A.

-> Suggestion: Gel filtration profiles of DrMoaE and DrMoaER110A.

-> Please give the calculated MW information with MW standard proteins.

Response 13: Thank you. It has been corrected (Lines 127, Page 4). The MW information has been shown in Figure 2D and Supplementary Figure 1.

Point 14: Throughout the manuscript, please use complementary or complementation instead of compensatory or compensation.

Response 14: Thank you for your suggestion. All of them have been corrected.

Point 15: Line 130

- in oxidation resistance

-> in oxidative stress resistance

Response 15: Thank you. It has been corrected (Line 145-146, Page 5).

Point 16: Figure 5 and Figure 6

-> Figures 5, 6 and its legends are not matched. Please clarify this and correct it adequately.

Response 16: Thank you for your reminder. The legends of Figure 5 and Figure 6 were incorrectly located. We have corrected this mistake (Line 188, Page 6; Line 219, Page 7).

Point 17: Line 183

- by affecting molybdenum enzyme activity.

-> Suggestion: by affecting DMSO reductase activity.

Response 17: Thank you. It has been improved (Line 200, Page 6).

Point 18: Lines 212-213

- In the present study, structural data of DrMoaE protein with a resolution of 1.9 Å were obtained using the diffraction technique.

-> In the present study, we solved the crystal structure of DrMoaE by 1.9 Å resolution.

Response 18: Thank you. It has been improved (Lines 229-230, Page 7).

Point 19: Throughout the manuscript, please clarify that ‘β2–β3 folds’ is proper term to express what authors mean.

Response 19: Thank you. It has been revised as “…...with a more disordered loop between β2 and β3 sheets……”(Line 282, Page 8).

Point 20: Line 273

- All D. r strains were

-> All D. radiodurans strains were

Response 20: Thank you. It has been corrected (Line 291, Page 8).

Point 21: Line 299-300

- at pool conditions

-> at a reservoir condition

Response 21: Thank you. It has been corrected (Lines 317-318, Page 9).

Point 22: Line 301

- 20% antifreeze

-> 20% cryoprotectant (please give detailed component)

Response 22: Thank you. It has been revised as “……20% cryoprotectant (0.2 M lithium sulfate, 0.1 M Tris pH 8.5, 22% PEG3350, and 20% (w/v) glycerol) were added ……” (Lines 319-320, Page 9).

Point 23: Line 303

- at the BL17U line station of

-> at the beamline BL17U of

Response 23: Thank you. It has been corrected (Line 322-323, Page 9).

Point 24: Line 305-309

- and modeled by molecular replacement using E. coli homologous structure MoaE (PDB:1NVJ). The refinement of the structure was used by PHENIX, and manual modeling was performed using COOT software. The refined structure included 139 amino acids of DrMoaE (residues 9-145 of each monomer), and the structure of DrMoaE was mainly presented using PyMOL software.

-> (reference for XDS program suit) and molecular replacement phasing using EcMoaE (PDB:1NVJ) as a search model was performed by using ~~(give MR program used in this study and its reference). Manual model building was performed using COOT software (add reference). The refined structure included 139 amino acids of DrMoaE (residues 9-145 of each monomer), and the structure of DrMoaE was mainly presented using PyMOL software.

Response 24: Thank you. It has been revised as “DrMoaE protein crystals were subjected to X-ray diffraction at the beamline BL17U of the Shanghai Light Source (SSRF) and were integrated and scaled using the XDS suite suit [36]. The structure of DrMoaE was solved by molecular replacement using  E. coli homolog (PDB:1NVJ; [23]) as the search model. The refinement of the structure was performed using PHENIX [37], and manual modeling was performed using COOT [38]. The refined structure included 139 amino acids of DrMoaE (residues 9-145 of each monomer), and the structure of DrMoaE was mainly presented using PyMOL” (Lines 323-331, Page 9).

Reference

  1. Rudolph, M.J.; Wuebbens, M.M.; Turque, O.; Rajagopalan, K.V.; Schindelin, H. Structural studies of molybdopterin synthase provide insights into its catalytic mechanism. J Biol Chem 2003, 278, 14514–14522, doi:10.1074/jbc.M300449200.
  2. Kabsch, W. Xds. Acta Crystallogr D Biol Crystallogr 2010, 66, 125–132, doi:10.1107/S0907444909047337.
  3. Adams, P.D.; Afonine, P.V.; Bunkoczi, G.; Chen, V.B.; Davis, I.W.; Echols, N.; Headd, J.J.; Hung, L.W.; Kapral, G.J.; Grosse-Kunstleve, R.W.; et al. PHENIX: a comprehensive Python-based system for macromolecular structure solution. Acta Crystallogr D Biol Crystallogr 2010, 66, 213–221, doi:10.1107/S0907444909052925.
  4. Emsley, P.; Lohkamp, B.; Scott, W.G.; Cowtan, K. Features and development of Coot. Acta Crystallogr D Biol Crystallogr 2010, 66, 486–501, doi:10.1107/S0907444910007493.

Point 25: Throughout the manuscript, please unify mL or ml.

Response 25: Thank you. All of them have been corrected.

Point 26: Lines 365-367

- One SOD enzyme activity unit (U) was defined when the inhibition percent of the reaction system was 50%.

-> For arbitrary unit definition, authors need to give more information such as the concentration of substrate, temperature, and pH and so on.

Response 26: Thank you. It has been revised as “Superoxide anion (O2-) that was produced by the reaction of xanthine and xanthine oxidase in the kit reduced nitro-blue tetrazolium to blue formazan. After the addition of SOD, O2- was cleared so as to inhibit the formation of formazan. It was an enzyme activity unit (U) when the percentage of SOD enzyme inhibition of the xanthine oxidase coupling reaction system described above was 50% at 37ºC” (Lines 389-394, Page 10-11).

Reviewer 2 Report

I suggest this manuscript IJMS-2106517 to be considered for publication after minor revision. Overall, the work is finished with decent quality. The authors structurally and functionally characterized MoaE from Deinococcus radiodurans both in vitro and in vivo. Experiments were performed scholarly. The entire logical flow of the manuscript can be improved because the structural study is not connecting with the functional studies.

1.    The authors need to provide additional data to support their claim for R110A mutant’s monomer behavior, for example, the exact elution volumes for both wild-type and R110A MoaE on the gel-filtration trace, as well as a calibration curve for the exact column used to calculate apparent molecular weight of both proteins.

Also, the authors failed to introduce with dimer formation is important to study and what if R110A is subjected to one of the assays shown in later figures, what the phenotype would be like. I request that the authors either discuss this or in a better way, generate the deletion strain of drmoaE that is compensated by R110A mutant, and test the phenotype.

2.    The color themes used in Fig3B and 3C are different versus Fig 3A and 3D, between the deletion drmoaE strain and the complementation strain. In my initial review of the figures, I believed UV and r-ray treatments’ results were unexpected and I had to go to the main text and the figure legends to follow the color theme used. Please the authors correct and use same color schemes for all 4 subpanels in this figure to help readers to understand more smoothly. Meanwhile, the r-ray treatment did not seem to generate a significant different phenotype between wild-type and deletion drmoaE strain though. I hope the authors can correct their language usage in the main text for this part.

3.    Fig5A needs to be color-themed graphed. The current version is difficult to follow between curves.

Author Response

Point 1: The authors need to provide additional data to support their claim for R110A mutant’s monomer behavior, for example, the exact elution volumes for both wild-type and R110A MoaE on the gel-filtration trace, as well as a calibration curve for the exact column used to calculate apparent molecular weight of both proteins.

Also, the authors failed to introduce with dimer formation is important to study and what if R110A is subjected to one of the assays shown in later figures, what the phenotype would be like. I request that the authors either discuss this or in a better way, generate the deletion strain of drmoaE that is compensated by R110A mutant, and test the phenotype.

Response 1: Thank you for your valuable suggestions. The calibration curve has been added in Figure 2D, and the phenotype results of ΔdrmoaE_R110A complementary strain have been shown Figure 3, respectively.

Point 2: The color themes used in Fig3B and 3C are different versus Fig 3A and 3D, between the deletion drmoaE strain and the complementation strain. In my initial review of the figures, I believed UV and r-ray treatments’ results were unexpected and I had to go to the main text and the figure legends to follow the color theme used. Please the authors correct and use same color schemes for all 4 subpanels in this figure to help readers to understand more smoothly. Meanwhile, the r-ray treatment did not seem to generate a significant different phenotype between wild-type and deletion drmoaE strain though. I hope the authors can correct their language usage in the main text for this part.

Response 2: Thank you so much. The color of Fig3 has been revised. Meanwhile, the information about γ-ray treatment has also has been improved (Line 139-141, Page 5).

Point 3: Fig5A needs to be color-themed graphed. The current version is difficult to follow between curves.

Response 3: Thank you for your suggestion. It has been revised.
